# Ultrashort laser pulse doubling by metal-halide perovskite multiple quantum wells

Jia Guo[1,6], Tanghao Liu[1,6], Mingjie Li [2,6✉], Chao Liang[1], Kaiyang Wang[1], Guo Hong[1], Yuxin Tang[1], Guankui Long[3], Siu-Fung Yu[2], Tae-Woo Lee [4], Wei Huang[5] & Guichuan Xing [1✉]

Multiple ultrashort laser pulses are widely used in optical spectroscopy, optoelectronic manipulation, optical imaging and optical signal processing etc. The laser pulse multiplication, so far, is solely realized by using the optical setups or devices to modify the output laser pulse from the optical gain medium. The employment of these external techniques is because the gain medium itself is incapable of modifying or multiplying the generated laser pulse. Herein, with single femtosecond laser pulse excitation, we achieve the double-pulsed stimulated emission with pulse duration of around 40 ps and pulse interval of around 70 ps from metal-halide perovskite multiple quantum wells. These unique stimulated emissions originate from one fast vertical and the other slow lateral high-efficiency carrier funneling from low-dimensional to high-dimensional quantum wells. Furthermore, such gain medium surprisingly possesses nearly Auger-free stimulated emission. These insights enable us a fresh approach to multiple the ultrashort laser pulse by gain medium.

[1] Joint Key Laboratory of the Ministry of Education, Institute of Applied Physics and Materials Engineering, University of Macau, 999078 Macao, China. [2] Department of Applied Physics, The Hong Kong Polytechnic University, Hung Hom, Kowloon, Hong Kong, P.R. China. [3] School of Materials Science and Engineering, National Institute for Advanced Materials, Nankai University, 300350 Tianjin, P.R. China. [4] Department of Materials Science and Engineering, Seoul National University (SNU), Seoul, Republic of Korea. [5] Institute of Flexible Electronics (IFE), Northwestern Polytechnical University (NPU), 710072 Xi'an, Shaanxi, P.R. China. [6] These authors contributed equally: Jia Guo, Tanghao Liu, Mingjie Li. ✉email: ming-jie.li@polyu.edu.hk; gcxing@um.edu.mo

Ultrashort laser pulse multiplication and shaping[1,2] have been used for many meaningful and unique applications, such as the optoelectronic manipulation[3,4], multiple-pulse laser plasma generation and ablation[5], optical imaging[6,7], optical communication[8,9], and nonlinear fiber optics[10], etc. At present, the double or multiple laser pulses generation from single ultra-short laser pulse is mainly achieved by using the delay lines and interferometer[11–13], multi-order wave plate and polarizer[14], birefringent crystal array[15,16], liquid crystal arrays[17], or integrated passive photonic devices[18], etc. All these techniques are using external optical or mechanical methods to modify the output laser pulses from the gain medium, which simulates our interest on whether there is a type of optical gain medium with unique optical properties and carrier dynamics that is capable of generating multiple laser pulses. Whereas, no such gain medium has yet been achieved in traditional materials due to the mandatory requirement of the high-efficiency multi-step energy relaxations.

Recently, solution-processed quasi-two-dimensional (quasi-2D) metal-halide perovskite multiple quantum wells (MQWs) with the structure of $L_2A_{n-1}M_nX_{3n+1}$ emerges as a promising candidate for the efficient light-emitting media[19–24]. Such perovskite comprises of van-der-Waals-coupled different thickness QWs with long organic ammonium cations (L) located between these slabs as spacers. Each QW, where the excitons are quantum confined, has an $n$-layer inorganic $[MX_6]$ octahedra with short monovalent organic cation A (e.g., $CH_3NH_3^+$) located at the void of the network for electrostatic charge-balancing. In such MQWs, the width of QW with different $n$-values is widely distributed. The thick high-dimensional (HD) QWs (e.g., QWs with $n$ larger than 5) with narrow bandgaps are surrounded by the host thin low-dimensional (LD) QWs (e.g., QWs with $n$ smaller than 5) with larger bandgaps (Fig. 1a). At low charge-carrier injection-density for electroluminescence (lower than $10^{16}$ cm$^{-3}$), efficient charge carrier transfer from LD perovskite QWs to HD ones has been observed in these MQWs perovskites (Fig. 1b)[19,20,24,25]. Such unique carrier funneling process would be favorable for achieving the population inversion in HD perovskite QWs that act as optical gain media for stimulated emission (StE). The gain dynamic processes could also be tailored by engineering the well width distribution and ordering in the MQWs film. It thus provides us a possible approach to modify the input ultrashort laser pulse number through tuning the StE dynamics. Nevertheless, to our best knowledge, the effects of this carrier funneling process on the optical gain dynamics have rarely been studied in such perovskite MQW system.

In this work, we demonstrate the sequential double pulsed StE from metal-halide perovskite MQWs film under the femtosecond (fs) laser excitation. The dual picosecond (ps) StE pulses with a time interval of around 70 ps originate from one fast and one slow charge carrier funneling channels from the LD QWs (dominated by $n = 2$ QWs) to the gain media of HD QWs (with $n$ larger than 5). Such unique carrier funneling processes also result in the nearly Auger-free StE. These insights may lead to the development of low-threshold ultrashort laser pulse doubling devices.

## Results

### Double-pulsed StE from perovskite QWs.
The perovskite MQWs film in this study consists of $(NMA)_2(FA)_{n-1}Pb_nI_{3n+1}$ (NMA = $C_{10}H_7CH_2NH_3$, FA = $NH_2CHNH_2$) with different inorganic layer

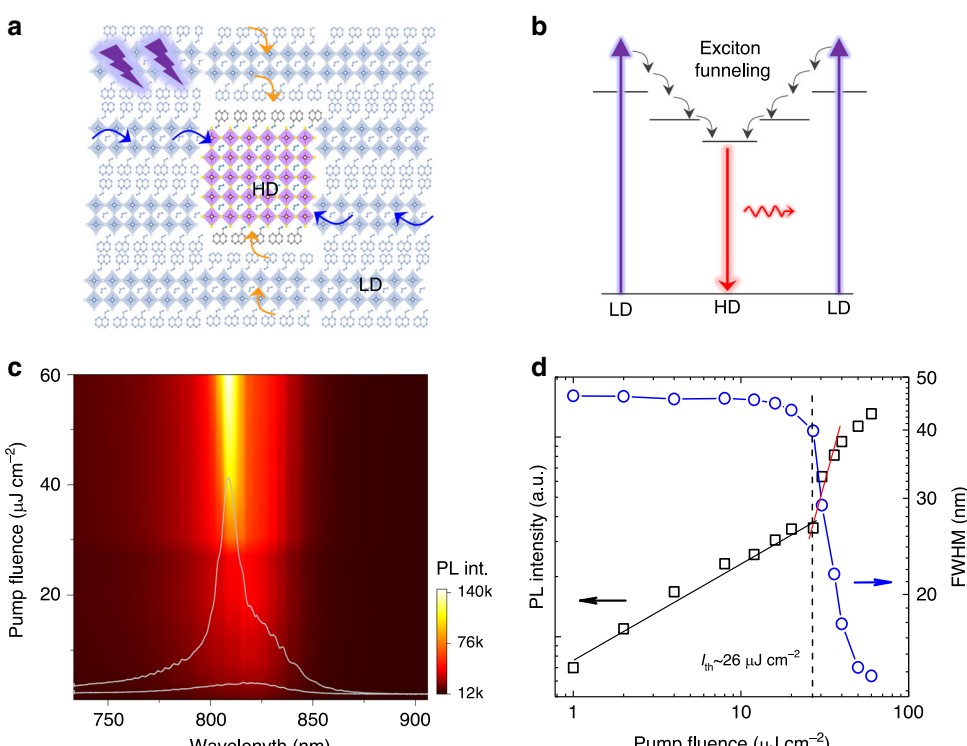

**Fig. 1 Stimulated emission from perovskite multiple quantum wells (MQWs). a** Schematic illustration of the localization of photogenerated charge carriers from the surrounding low-dimensional (LD, dominated by $n = 2$ QWs) to high-dimensional (HD, $n \geq 5$) perovskite QWs. The blue and orange arrows indicate the lateral and vertical charge transfer, respectively. **b** Schematic energy level diagram shows the photogenerated excitons funneling process. **c** 2D pseudocolor plot of the photoluminescence (PL) intensities of perovskite MQWs with pump fluence from 1 to 60 μJ cm$^{-2}$ (under 400 nm, 50 fs, 1 kHz fs laser excitation). The broad and sharp curves represent the PL spectra below and above the stimulated emission threshold, respectively. **d** Log–log plot of the integrated PL intensities and the full width at half maximum (FWHM) as a function of pump fluence. The black and red solid lines are linear fit slopes of PL intensities below and above the stimulated emission threshold.

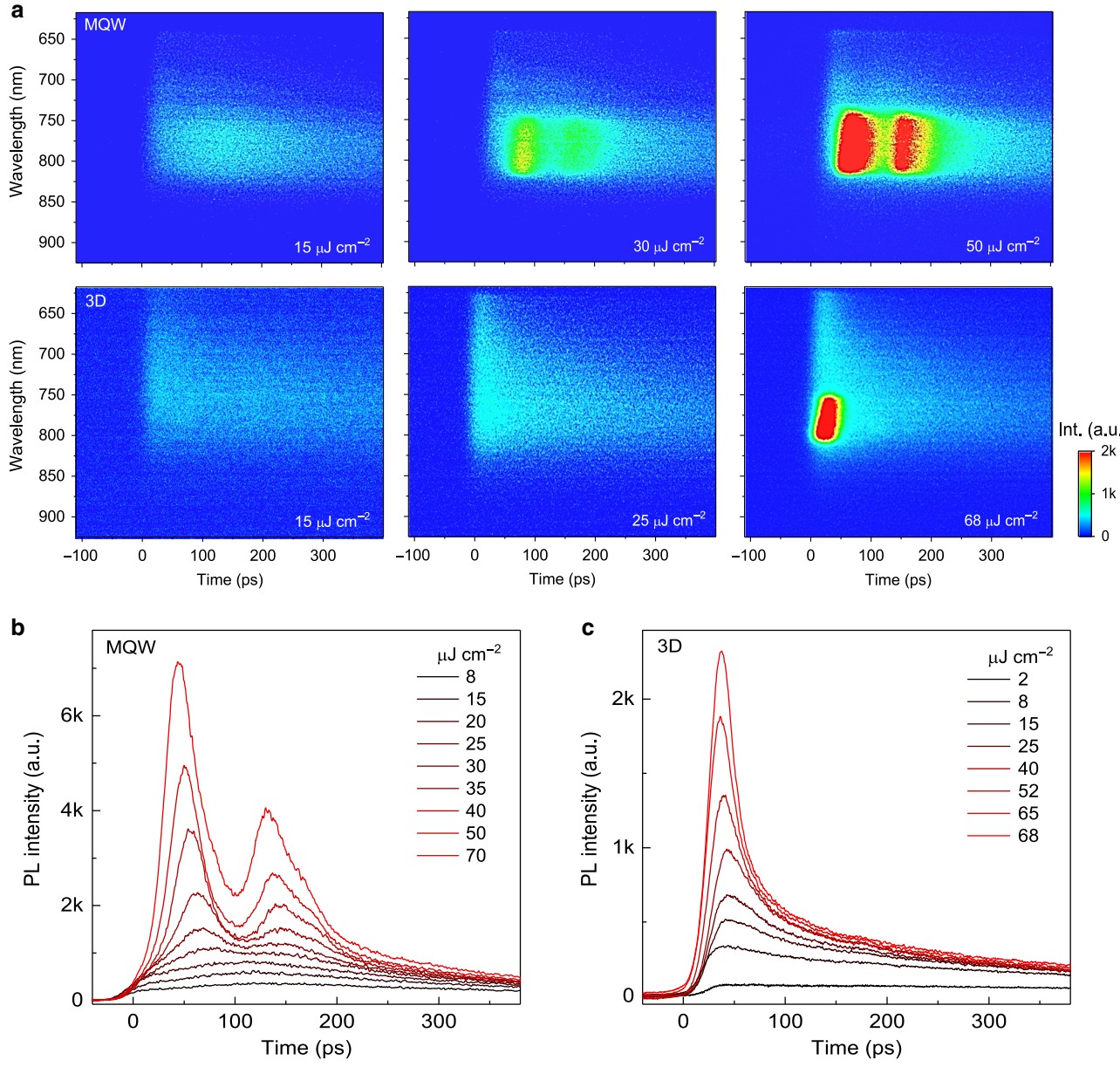

**Fig. 2 Ultrashort laser pulse doubling by perovskite multiple quantum wells (MQWs). a** Pseudocolor time-resolved photoluminescence (TRPL) images show the spectrum evolution with time of the MQWs (upper panel) and 3D (lower panel) perovskite films at three representative pump fluences, respectively. **b** and **c** The extracted TRPL decay curves of **b** perovskite MQWs at the position of 780 ± 20 nm and **c** 3D perovskite at the position of 790 ± 20 nm under different pump fluences. These results were collected from samples excited with 400-nm laser pulses (1 kHz, 50 fs).

numbers ($n$) as determined by the crystal structure and UV–Vis characterizations (Supplementary Fig. 1). The dominant component is 2D $(NMA)_2(FAPbI_3)PbI_4$ bilayer QWs ($n = 2$). These QWs are weakly assembled by van-der-Waals force with atomically sharp interfaces. To characterize the optical gain behavior, the film sample is pumped by a 400-nm laser beam (around 50 fs pulse width, 1 kHz repetition rate) with a stripe spot dimension of 5 mm × 0.2 mm focused by a cylindrical lens to form a waveguide. The 400-nm laser pulse would mainly excite the dominant LD bilayer QWs. The emission is collected at the cleaved side surface of the film. At low pump fluence, the photoluminescence (PL) is dominated with a broad spontaneous emission centered at around 800 nm, which originates from the HD perovskite QWs. With increasing pump fluence above the threshold pump fluence ($I_{th}$) of around 26 μJ cm$^{-2}$, the PL intensity increases sharply together with bandwidth collapses from larger than 40 nm to around 10 nm

(Fig. 1c and d). The slope of the power-dependent output PL intensity increases from 0.5 below the threshold to around 3 above the threshold and finally approaches around 1 at high pump fluence, corresponding to the transitions from spontaneous emission to amplification by StE and ultimately to gain saturation[26,27]. These features clearly indicate that the HD QWs serve as the optical gain medium. In comparison, the threshold of StE from the control sample, 3D FAPbI$_3$ perovskite film, is around 45 μJ cm$^{-2}$ under the same excitation conditions (Supplementary Fig. 2).

Time-resolved PL (TRPL) is measured to monitor the StE dynamics in the gain media by using a streak camera with an ultimate time resolution of around 20 ps. The upper panel of Fig. 2a, from left to right, shows the spectra time-evolution of the perovskite MQWs film under different pump fluences. Below the StE threshold fluence, the transient emission intensity increases gradually with a peak at around 150 ps (Supplementary Fig. 3),

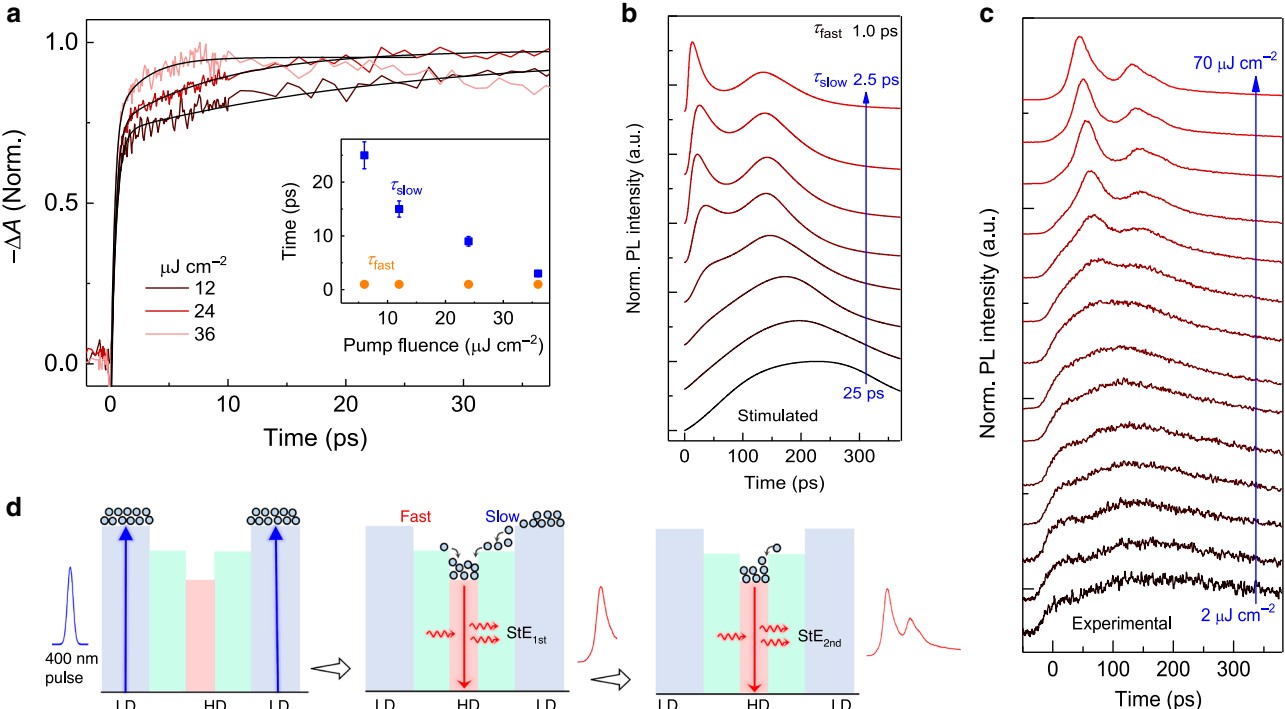

**Fig. 3 Origin of the ultrashort laser pulse doubling. a** Normalized transient absorption (TA) dynamics probed at the excitonic photobleaching peak (at 750 nm) of the high-dimensional (HD, $n \geq 5$) perovskite QWs in the gain medium. Inset shows the fitted TA peak rising time as a function of pump fluence. The error bars represent the uncertainties of the fittings. **b** Simulated carrier recombination dynamics at different carrier funneling times and recombination rates in the perovskite MQWs gain medium. See Supplementary Note 1 and Supplementary Table 1 for detailed parameter values. **c** Normalized time-resolved photoluminescence (TRPL) decay curves for perovskite MQWs from low to high pump fluence following excitation at 400 nm (1 kHz, 50 fs). **d** Schematic illustration of the formation of double-pulsed stimulated emission (StE) by single fs laser pumping via the fast and slow carrier funneling processes in MQWs.

which was attributed to the charge carrier funneling from LD to HD perovskite QWs[20]. Interestingly, when the pump fluence is increased to above the StE threshold $I_{th}$, two separated transient StE pulses with peak widths of around 40 ps and interval of around 70 ps appear as reflected by the two bright spots in streak camera images (Fig. 2a, upper panel) and two peaks in TRPL decay curves (Fig. 2b and Supplementary Fig. 3 for log plots). The fitted decay lifetimes of first and second StE peaks are 33 and 50 ps, respectively (Supplementary Fig. 4), which are much shorter than that of the spontaneous emission before the StE. These results thus indicate the observation of the double-pulsed StE from the perovskite MQWs after single 400-nm fs laser pulse pumping. In another word, the metal-halide perovskite MQWs as a gain medium is able to double the input ultrashort laser pulse.

In contrast, for the conventional optical gain medium of 3D perovskite film, no delayed building-up processes of the spontaneous emission is observed (Fig. 2c and Supplementary Fig. 3). For such normal optical gain medium with uniform dimensionality, all the photoexcited charge carriers are directly injected into the optical gain excited levels within laser pulse duration. These initially excited hot-carriers then relax rapidly to the band edge via carrier–phonon interaction, followed by the recombination (see the schematic process in Supplementary Fig. 5). At higher pump fluence above the threshold for population inversion, one StE pulse with a lifetime of around 20 ps (limited by the time resolution of streak camera) is observed (Fig. 2a, lower panel), which is consistent with generally observed StE dynamics in most of the traditional optical gain media[28–30].

Moreover, to double-confirm that the observed dual-pulsed StE is related with the carrier funneling processes from LD to the HD QWs, the 650-nm laser pulse is used to only excite the HD

perovskite QWs in the MQWs film with the same pumping configuration used for 400-nm laser excitation. Similar to the 3D perovskite, one StE pulse is observed (Supplementary Figs. 5 and 6) with a threshold pump fluence of 37 μJ cm$^{-2}$. These observations rule out other possible origins for double-pulsed StE, such as the multiple reflections in the sample[31], which should not be pumping photon energy-dependent. Besides, the exciton–lattice interaction in 2D perovskite QWs[32,33] is unlikely to be the origin of double-pulsed StE as no oscillations are observed in our TRPL measurements at different pump fluences. Thus, the existence of the unique charge carrier funneling processes from the initially excited bilayer LD to the HD QWs in the perovskite MQWs is the key to achieve the double pulsed StE.

**Two-step carrier funneling processes to gain medium.** To investigate the mechanism of the observed double-pulsed StEs in perovskite MQWs, we monitored the carrier funneling processes to the HD QWs using transient absorption (TA) spectroscopy. Upon 400-nm laser pulse excitation, the charge carriers are primarily injected into the bilayer QWs given their dominated light absorption at this wavelength and limited HD QW domain concentration (estimated to be around $9 \times 10^{16}$ cm$^{-3}$) within the MQWs. Due to the smaller bandgap of HD QWs, the photo-injected carriers will funnel from LD QWs to HD QWs, competing with the carrier recombination within the LD QWs. Such kind of carrier funneling in 2D heterojunctions could be extremely fast and efficient for the atomically flat interface and quantum coupling between the adjacent 2D layers[34–37]. To monitor the carrier funneling process, in TA spectroscopy, the photoinduced absorption change ($\Delta A$) of the MQWs film is probed with a time-delayed white light pulse (Supplementary

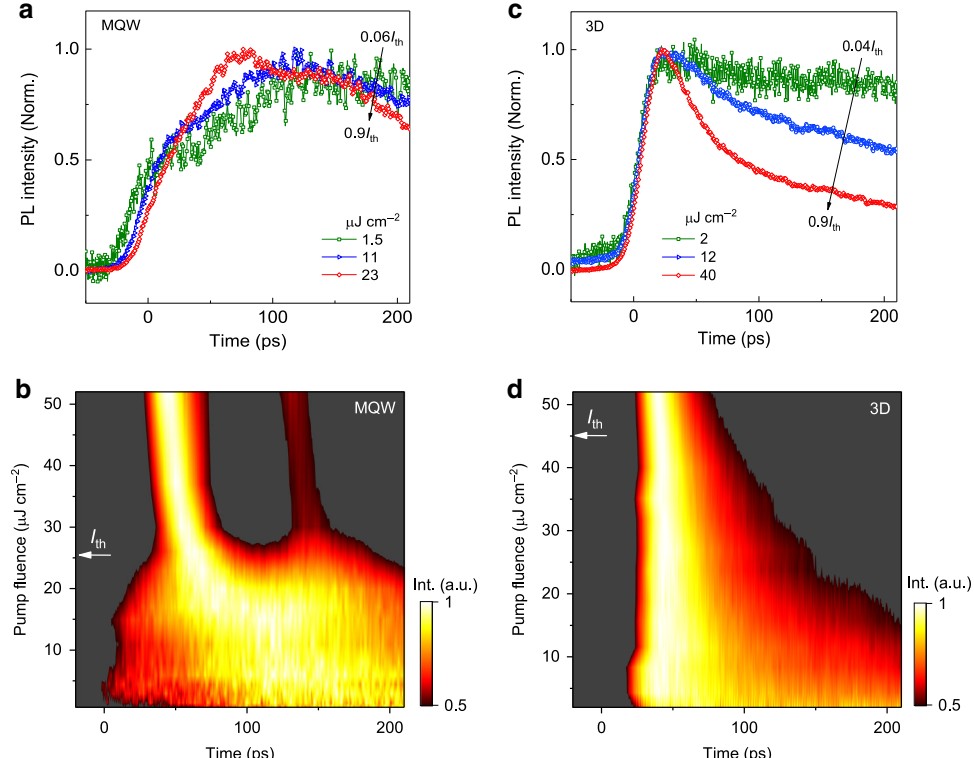

**Fig. 4 Nearly Auger-free stimulated emission (StE). a** Normalized time-resolved photoluminescence (TRPL) decay curves of the perovskite MQWs film pumped below the StE threshold fluence ($I_{th}$) of 26 μJ cm⁻². **b** 2D pseudocolor plot of normalized TRPL intensity monitored at the emission peak of gain transitions in MQWs film under different pump fluences. **c** Normalized TRPL decay curves of the 3D perovskite film pumped below the StE $I_{th}$ of 45 μJ cm⁻². **d** 2D pseudocolor plot of normalized TRPL intensity monitored at the emission peak of gain transitions in 3D perovskite film under different pump fluences. The white arrows indicate the position of the threshold pump fluence of StE.

Fig. 7). The TA spectra of MQWs film consist of a dominant bleaching peak from $n = 2$ LD QWs (at 570 nm) and a broad bleaching band from $n \geq 5$ QWs (with a peak at 750 nm, the energy position of gain medium) accompanied by several very weak peaks corresponding to $n = 3$ and 4 QWs. The carrier population in different dimensional QWs can be monitored from the TA bleaching dynamics at their respective exciton transitions. Figure 3a shows the TA dynamics probed at the peak of the gain transitions (i.e., $n \geq 5$ QWs) of MQWs. The fast decay channel possesses a near-constant rising time of $0.8 \pm 0.2$ ps, while the relatively slow rising channel possesses a lifetime decreasing from 25 to 2.5 ps with increasing the pump fluence from 6 to 37 μJ cm⁻² (Fig. 3a inset). These results clearly demonstrate a two-step growing dynamics at the gain transitions, which is also closely matched with the two-step carrier depopulation at the bilayer QWs (Supplementary Fig. 7d). Therefore, the initially injected-carriers efficiently funnel from the dominant bilayer QWs into the optical gain states via two decay channels. The TRPL of optical gain medium with one fast and another slow-growing PL dynamics before StE further confirm such two-step carrier funneling process (Fig. 3c). In contrast, in 3D perovskite, the rising time of TA signal probed at the bleaching peak of gain medium (at 760 nm) slightly increases with increasing the pump fluence (Supplementary Fig. 8), which is ascribed to the hot-phonon-induced slower hot-carrier cooling lifetime at higher carrier densities[38–40]. Considering the fact that the large lateral to vertical dimension ratio (aspect ratio) of LD QWs in MQWs, the relatively slow carrier funneling channel could originate from the lateral charge carrier migration within the LD QWs and then transfer to the HD QWs (as indicated with blue arrows in Fig. 1a). And the fast carrier funneling channel would be the

vertical charge transfer across the stacked QWs via quantum coupling between each other (orange arrows in Fig. 1a). With the limited HD QW domain concentration in the MQWs, at higher pump fluence, more energy states of HD QWs can be occupied quickly. Therefore, the lifetime of the relatively slow carrier funneling channel to the HD QWs will become shorter at higher pump fluence due to the population saturation of the HD QWs.

**Double-pulsed StE mechanism.** Here, the dual-pulsed StE could be modeled via the rate equations with two charge-carrier funneling channels from the dominant bilayer LD QWs to the HD QWs (i.e., $n \geq 5$ QWs). Based on the above TA dynamic findings, in the simulation, we applied one carrier funneling process with a fixed fast lifetime of 1 ps from charge transfer of stacked LD QWs, and another lateral charge transfer with variable lifetime from 25 to 2.5 ps with increasing excitation level (see Supplementary Fig. 9 and Supplementary Note 1 for details). The excitation level is tuned by changing the population probabilities of excited state levels. Figure 3b shows the simulated recombination dynamics from the gain states at different excitation levels. The emission is governed by a slow rising process at low excitation level, followed by the appearance of the two fast emission pulses at a high-excitation level, which is consistent with the observed emission dynamics (see the normalized TRPL in Fig. 3c). Thus, these results suggest the prerequisite effects of the two carrier funneling channels on the generation of such dual-pulsed StE (Fig. 3d). The excitons are first generated in the dominant bilayer QWs by the 400-nm laser pulse. When excitons continuously localize to the gain states (i.e., $n \geq 5$ QWs), the optical gain increases, the StE occurs once the optical gain is

larger than the gain losses in the gain medium. The first StE occurs within around 70 ps after excitation, and the localization of the rest excitons from another channel results in the second StE appeared at around 140 ps. It should be noted that the fast carrier transfer bypasses the nonradiative trapping centers in the LD perovskite QWs with the estimated efficiency as high as over 85% by the PL and TA measurements. Such efficient carrier transfer from LD to HD QWs, therefore, ensure the rest excitons can be safely transferred to the HD perovskite QWs for the building-up of the second population inversion and the sub-sequent StE pulse. The weak shoulder on the right of the second StE pulse at around 170 ps (Fig. 2b) may originate from the carrier funneling from some farther LD QWs to the gain medium after the second StE, and the weaker intensity may be due to the limited amount of such LD QWs.

**Nearly Auger-free-StE**. To evaluate the efficiency of such double-pulsed StE, we compared the emission dynamics in perovskite MQWs and normal 3D perovskite films on the short dynamic time range, which mainly contains the StE and nonradiative Auger recombination processes. Surprisingly, for perovskite MQWs, the emission lifetimes within around 200 ps under the pump fluence increased from $0.06I_{th}$ to $0.9I_{th}$ are almost unchanged (as shown in TRPL curves in Fig. 4a, and the time-fluence and pump-fluence-dependent emission intensities in Fig. 4b), which indicates a nearly direct transition from the spontaneous emission to StE with an absence of the typically observed emission-lifetime gradual shortening. In contrast, for the 3D perovskite, the emission lifetime decreased significantly from longer than 1 ns to around 25 ps with increasing the pump fluence from $0.04I_{th}$ to $0.9I_{th}$ (Fig. 4c and d), which indicates the severe Auger recombination as well as the accelerated bimolecular recombination[41]. As the optical gain (proportional to carrier density) has to compete with such losses induced by fast decay of carriers, in 3D perovskite, a higher threshold pump fluence is therefore needed for StE. The observed nearly unchanged emission dynamics in perovskite MQWs, therefore, implies the nearly Auger-free StE, which was rarely observed in traditional single-dimensional optical gain media (for example, including 2D perovskites[42], 3D perovskites films and nanostructures[28,43,44], colloidal III–V and II–VI semiconductor quantum wells and quantum dots[29,45–48]). The spatially separated energy level system in the perovskite MQWs could store the excited carrier energies in higher energy levels of LD QWs. Thus, the carrier concentration in HD QWs is gradually increased due to the carrier funneling rather than the instantaneous carrier injection in the optical gain states, which is beneficial for a reduced energy loss by suppressed Auger recombination and decreased StE threshold pump fluence. As shown in Fig. 4b, the StE peaks continuously redshift to the earlier time with increasing pump fluence (see the peak positions as a function of pump fluence in Supplementary Fig. 10). Since the rising time of StE represents the optical gain building up time, and it inversely depends on the gain coefficient. As the optical gain coefficient continuously increases with increasing carrier concentration, when the carriers from LD QWs funnel into the HD QWs, the gain coefficient will gradually increase, and consequently the building up time of StE pulses decreases at the high pump fluence.

Furthermore, we found that the pulse interval of two transient StE peaks slightly increases from around 70 to 87 ps with increasing the pump fluence from 26 to 70 μJ cm⁻². Such an increase in pulse interval is due to the faster building up of the first StE pulse with increasing the pump fluence (Supplementary Fig. 10). At higher pumping fluence, more carriers could be directly injected into the HD QWs gain medium, which thus induces the shorter optical gain building up time for the first StE pulse than that of the second one from the carrier funneling. Except for the carrier concentration, the pulse interval could be further controlled through tuning the carrier funneling time via the following measures such as changing the organic spacers[49], the content ratios of LD and HD QWs or crystallization engineering[50] to modify the carrier funneling efficiency or rate in perovskites MQWs, which need to be investigated in the future studies.

## Discussion

In summary, the ultrashort laser pulse doubling is observed in fs laser-pulsed excited metal-halide perovskite MQWs. Such unique behavior is due to the efficient two-step carrier funneling processes in the mixed-dimensional MQWs acting as the optical gain medium. In the MQWs stacked together with atomically flat interfaces and relatively weak van-der-Waals force, the gain dynamics is related to one fast vertical and another relatively slow lateral efficient carrier funneling from the low to the high-dimensional perovskite QWs. The observed dual-pulsed StE could be well modeled with the two-step carrier funneling processes. Such carrier funneling processes also result in the suppressed Auger recombination. These unique properties distinguish metal-halide perovskite MQWs as a gain media favorable for developing low-threshold ultrashort laser pulse doubling devices.

## Methods

**Materials preparation**. The $C_{10}H_7CH_2NH_3I$ (NMAI) is synthesized by dropping stoichiometric amount of hydroiodic acid (4.34 g, 45 wt% in water) into 1-naphthalenemethylamine (12.72 mmol in 50 mL tetrahydrofuran (THF)) at 0 °C and stirring condition for 2 h. Then the solution is removed with rotary eva-poration at 50 °C. The remains are washed three times with THF: $CH_2Cl_2$ (3:1 v/v) solution and then dried under vacuum at a temperature of 60 °C overnight. The precursors of NMAI, $NH_2CHNH_2I$ (FAI), and $PbI_2$ with a molar ratio of 2:1:2 are dissolved in DMF solution with $Pb^{2+}$ molar concentration of 0.2 M. After rigorous stirring at 60 °C for 1 h in a nitrogen atmosphere, the multiple quantum well perovskite films are deposited by spin coating this solution onto quartz substrates, followed by annealing at 100 °C for 10 min. The 3D $FAPbI_3$ films are deposited with the same method by dissolving stoichiometric amount of FAI and $PbI_2$ in DMF with $Pb^{2+}$ molar concentration of 0.2 M.

**Time-integrated PL and TRPL spectroscopy**. Time-integrated steady-state PL from the samples was collected at the edge of the film by using a pair of lenses into an optical fiber that was coupled to a spectrometer (Acton, Spectra Pro 2500i) and detected by a charge-coupled device (Princeton Instruments, Pixis 400B). TRPL was measured using an Optronis Optoscope streak camera system.

**TA spectroscopy**. The broadband fs TA spectra were collected by using a TA spectrometer (HELIOS, Ultrafast System). The white light probe beam (420–800 nm) was generated by focusing a small portion (around 10 μJ) of the fundamental 800 nm laser pulses into a 2 mm sapphire plate. The 400-nm pump pulses were obtained through doubling the fundamental 800 nm pulses with a BBO crystal and focused on the sample with beam size of 2 mm in diameter.

**Photoexcitation laser source**. The 400-nm pump laser for PL, TRPL, and TA spectroscopies were generated from the frequency doubling the 800-nm funda-mental regenerative amplifier output using a thin BBO crystal. The 800-nm fun-damental pulse with pulse width of around 150 fs is from a 1 kHz regenerative amplifier and was seeded by a mode-locked Ti–sapphire oscillator. The 650-nm pump laser is obtained from an optical parametric amplifier (Coherent OPerA Solo) pumped by the regenerative amplifier. All optical measurements were per-formed at room temperature with samples in nitrogen atmosphere or vacuum.

## Data availability

The data supporting the plots within this paper and other findings of this study are available from the corresponding authors upon request.

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

## Acknowledgements

We thank Professor T.C. Sum from Nanyang Technological University for providing some experimental facilities as well as for scientific discussions. The authors are grateful to the start-up fund (P0031122) from the Hong Kong Polytechnic University, the Macau Science and Technology Development Funds (FDCT-116/2016/A3, FDCT-091/2017/A2, and 014/2017/AMJ), the Research Funds (SRG2016-00087-FST and MYRG2018-00148-IAPME) from University of Macau and the Natural Science Foundation of China (61605073, 61935017, and 91733302) for financial support for this work.

## Author contributions

J.G., T.L., M.L., and G.X. conceived the idea, designed the experiments, and drafted the manuscript. J.G. and G.X. developed the basic concepts, conducted the spectroscopic characterization, and coordinated the experiments. J.G., T.L., and W.H. fabricated and characterized the samples. C.L., K.W., G.H., Y.T., G.L., S.-F.Y. and T.-W.L. contributed to the data analysis and manuscript preparation. M.L. and G.X. led the project.

## Competing interests

The authors declare no competing interests.
