## [Peer Review File · Nature Communications]

Reviewers' comments:

Reviewer #1 (Remarks to the Author):

In the manuscript of "ultrafast coherent light pulse doubling by metal-halide perovskite multiple quantum wells", Guo et al have studied the double optical pulse in perovskite multiple quantum wells (MQW). With the excitation at 400 nm, the authors have observed two pulse with 40 ps pulse duration and 90 ps interval. The authors have also carefully discussed the underlying mechanism for the formation of double pulses. A nearly Auger free process has also been revealed. These results are supposed to be very important for the low threshold perovskite microlasers. Overall, this research is very interesting and can be considered for publish after addressing the following comments.

1. In case of conventional multiple pulse lasers, the pulse interval can be precisely controlled. How the authors control their pulse interval? This should at least be discussed.
2. The authors have discussed the mechanism for two kinds of two-dimensional perovskites. One is $n < 5$ and the other has $n > 5$. This is relatively rough for a model since quantum wells with $n = 1$ should definitely be different from $n = 4$. The emission with $n = 5$ is also different from 3D perovskites. The authors should state clearly in the main text that the dominant layer number. This will help understanding the physics.
3. In page-7, the authors have compared the pumping with 650 nm laser. For this experiment, the threshold information should also be added in the main text. This can be an additional information of the energy transfer.
4. Figure 2b and c are too crowded. The pumping density is missing.
5. The same problem happens in Figure 4 a and c. Pumping density should be added. For this figure, the authors should consider a different plot since it is almost impossible to distinguish the data.
6. The reference number is too few. More references should be provided.
7. Please plot the data in Figure 1d in logscale to identify the gain of StEs.

Reviewer #2 (Remarks to the Author):

In this manuscript, the authors reported the observation of clearly resolved dual pulsed stimulated emission in perovskite multiple quantum wells. The mechanism of double-pulsed stimulated emission has been investigated in detail by using transient spectroscopy measurements and theoretical modelling. Furthermore, the Auger recombination in the perovskite multiple quantum wells in this work behaves obviously different from the normal gain medium. These observations of the ultrashort laser pulse doubling by using optical gain medium are novel and very interesting. The conclusions are convincing, and the manuscript is well-written. I believe this paper will attract extensive attentions in the field of metal-halide perovskites and ultrafast photonics. Therefore, I recommend the publication of this manuscript in Nature Communications.

I have few minor suggestions listed below and hope the authors would consider before publication:

1. There is very little discussion about previous works of the light-induced lattice dynamics on the stimulated emission in perovskites. The coherent acoustic phonon was observed in 2D perovskites [Nat. Commun. 9, 2019 (2018)]. How do the authors reconcile their observations with this previous report?
2. Can the authors rule out other possible reasons for the double pulsed stimulated emission? For example, the multiple reflection of the stimulated emission pulse within the film.
3. Do the authors observe any sample degradation under these conditions, and could this influence the overall carrier dynamics?
4. Can the authors provide more explanation for the shoulder on the right of the second pulse in Fig. 2b at ~ 170 ps at high pump fluence. Is it from the third stimulated emission pulse, why it is much weaker than second pulse?

Reviewer #3 (Remarks to the Author):

The authors of this manuscript present a study on the amplified spontaneous emission (ASE) in 2D perovskites. The particular structure of the material allows for a charge carrier reservoir, such that, after an initial ASE peak, a second peak is observed at longer time delay, due to funneling of carriers from the reservoir into the emitting state on a picosecond time scale.

Notwithstanding the technical merits of the manuscript, I do have the following two major concerns:

1. The authors are overselling their study.

-They study concerns the observation of ASE with two time constants. This is not equal to coherent light pulse doubling, at no point the coherence of the light pulses is demonstrated.

-They also claim to produce ultrashort pulses, yet the pulsed produced are more than two orders of magnitude longer than the 150 fs excitation pulses.

2. They also state that their results now enable a miniaturization of devices capable of light pulse doubling. However, on-chip manipulation of light pulses has already been demonstrated in silicon integrated photonics. Typically passive control of the light pulses is achieved, for instance, by microring resonators or photonic crystals acting as an optical delay line (e.g. doi: 10.1109/JPHOT.2010.2044989). Combined with waveguide splitters, one could be able to achieve a pulse train with controllable delay between pulses, in essence what the researchers aim to show here but with a higher degree of control.

In conclusion, considering the elements above, I find the manuscript more suitable for a more specialized journal, as it doesn't meet the stringent criteria of novelty and potential impact that merits publication in Nature Communications.

- All reviewer comments are displayed in *dark orange italics*.
- All our responses are displayed in black.
- Sentences indicating changes to the manuscript are **blue**.

Reviewer #1:

In the manuscript of "ultrafast coherent light pulse doubling by metal-halide perovskite multiple quantum wells", Guo et al have studied the double optical pulse in perovskite multiple quantum wells (MQW). With the excitation at 400 nm, the authors have observed two pulse with 40 ps pulse duration and 90 ps interval. The authors have also carefully discussed the underlying mechanism for the formation of double pulses. A nearly Auger free process has also been revealed. These results are supposed to be very important for the low threshold perovskite microlasers. Overall, this research is very interesting and can be considered for publish after addressing the following comments.

Response: We are delighted to hear the reviewer's recognition on the novelty and importance of our research discovery and greatly appreciate his/her recommendation for publication of our work. The reviewer's critical comments are very helpful for us to further improve the manuscript. We have carefully considered the reviewer's comments and made the necessary revision as follows:

1. In case of conventional multiple pulse lasers, the pulse interval can be precisely controlled. How the authors control their pulse interval? This should at least be discussed.

Response: We thank the reviewer for the constructive comment. Currently, it is extremely challenging to control the pulse interval by the gain medium itself. Because the pulse interval is dependent on the charge transfer time from low dimensional perovskite QWs, and the charge transfer rate is dependent on the carrier concentration, mobility, diffusion coefficient etc. inside the material. Possible ways to change the pulse interval may include purposeful engineering of the perovskite materials, such as changing the organic spacer between QWs, the cations, the QW thickness/dimensions, low-dimensional (LD) and high-dimensional (HD) QW content ratios in the perovskite MQWs, etc. Given the control or tuning of the pulse interval needs large amount of trial attempts, which is beyond the scope of this work, and shall be investigated in detail in the future studies.

Furthermore, although we have not demonstrated the fine tuning of pulse interval in our current manuscript, we do observe the variation of pulse interval along with the pump fluence as shown below:

Fig. R1. | Variation of the StE pulse intervals. **a** Pump-fluence dependent StE pulses positions. **b** Pump-fluence dependent intervals of two transient StE pulses. The error bars represent the uncertainties in the determination of StE pulse peak positions.

We therefore have included the following explanation of the pulse interval variation and discussed about the possible ways to tune the pulse intervals in our revised manuscript, and have also added the Fig. R1 as Supplementary Fig. 10 in the SI.

“Furthermore, we found that the pulse interval of two transient StE peaks slightly increases from ~ 70 ps to 87 ps with increasing the pump fluence. Such increase of pulse interval is due to the faster building up of the first StE pulse with increasing the pump fluence (Supplementary Fig. 10). At higher pumping fluence, more carriers could be directly injected into the HD QWs gain medium, which thus induce the shorter optical gain building up time for the first StE pulse than that of the second one from the carrier funneling. Except for the carrier concentration, the pulse interval could be further controlled through tuning the carrier funneling time via the following measures such as changing the organic spacers⁴⁷, the content ratios of LD and HD QWs or crystallization engineering⁴⁸ to modify the carrier funneling efficiency/rate in perovskites MQWs, which need to be investigated in the future studies.” on Page 15 of the revised manuscript.

2. The authors have discussed the mechanism for two kinds of two-dimensional perovskites. One is $n < 5$ and the other has $n > 5$. This is relatively rough for a model since quantum wells with $n = 1$ should definitely be different from $n = 4$. The emission with $n = 5$ is also different from 3D perovskites. The authors should state clearly in the main text that the dominant layer number. This will help understanding the physics.

Response: We thank the reviewer for the constructive comment.

Based on our UV-Vis (supplementary Fig. 1) and transient absorption (TA) measurements (supplementary Fig. 7) of the MQW sample, the dominant LD QWs in the sample are $n = 2$ QWs and the dominant HD QWs are $n \geq 5$ QW. The other QWs (i.e., $n = 3$, $n = 4$) are also observed from TA spectra but with very weak intensities. The corresponding values of n have been clearly labelled in the figure (supplementary Fig. 7).

Thus we have mentioned “The dominant component is the $(\text{NMA})_2(\text{FAPbI}_3)\text{PbI}_4$ bilayer QWs ($n = 2$).” on page 4 of the main text. And also mentioned “the charge carriers are primarily injected into the bilayer QWs due to their dominated light absorption” on page 9 of the main text.

As shown in TA spectra (supplementary Fig. 7), for the high-dimensional QWs, the bandgaps of QWs with different number of n are very close, it is hard to distinguish the QW dimensions when $n \geq 5$. All these QWs together contribute to a broad bleaching band (with peak at ~ 750 nm), which is also the energy position of gain medium. Therefore, we mentioned in the main text that “Figure 3a shows the TA dynamics probed at the gain transitions (i.e., $n \geq 5$ QWs) of MQWs.”

To make our statement clearer to readers, we made following changes in the revised manuscript:

(i) added “The TA spectra of MQWs film consist of a dominant bleaching peak from $n=2$ LD QWs (at ~ 570 nm) and a broad bleaching band from $n \geq 5$ QWs (with peak at ~ 750 nm, the energy position of gain medium) accompanied by several weak peaks corresponding to $n=3$ & 4 QWs.” on page 9 of main text.

(ii) “The excitons are first generated in the LD perovskites QWs by 400-nm laser pulse.” has been changed into

“The excitons are first generated in the **dominant bilayer** QWs by 400-nm laser pulse.” on page 12 of main text.

(iii) “Here, the dual pulsed StE could be modelled via the rate equations with two charge-carrier funneling channels.” has been changed into

“Here, the dual pulsed StE could be modelled via the rate equations with two charge-carrier funneling channels **from the dominant bilayer LD QWs to the HD QWs (i.e., $n \geq 5$ QWs)**.” on page 11 of main text.

3. In page-7, the authors have compared the pumping with 650 nm laser. For this experiment, the threshold information should also be added in the main text. This can be an additional information for the energy transfer.

Response: We thank the reviewer for the constructive comment.

We have revised the manuscript accordingly and added the pump fluence threshold as “Similar to the 3D perovskite, only one StE pulse is observed (see Supplementary Figs. 5&6) **with a threshold pump fluence of $37 \mu\text{J cm}^{-2}$** .” on Page 7 of the revised main text.

4. Figure 2b and c are too crowd. The pumping density is missing.

Response: We thank the reviewer for the constructive comment.

Following the reviewer’s suggestion, we have removed several TRPL curves and added the pump fluence accordingly in the revised Fig 2b and c. The TRPL curves with complete pump fluence are provided in Supplementary Figs. 3a and 3b for reference.

5. The same problem happens in Figure 4 a and c. Pumping density should be added. For this figure, the authors should consider a different plot since it is almost impossible to distinguish the data.

Response: We thank the reviewer for the constructive comment.

Following the reviewer's suggestion, we have re-plotted the Fig. 4a and c, and added the pump fluence accordingly.

6. The reference number is too few. More references should be provided.

Response: We thank the reviewer for the constructive comment.

Following the reviewer's suggestion, we have added 19 relevant reference in the revised manuscript.

7. Please plot the data in Figure 1d in logscale to identify the gain of StEs.

Response: Again, we thank the reviewer for the constructive comment.

Following the reviewer's suggestion, we re-plotted Fig. 1d in log-log scale. We also re-plotted Supplementary Figs. 2b and 6b in log-log scale for consistence and added the relevant discussion and references: “The slope of the power-dependent output PL intensity increases from 0.5 below the threshold to ~3 above the threshold and finally approaches ~1 at high pump fluence, corresponding to the transitions from spontaneous emission to amplification by StE and ultimately to gain saturation^{26,27}.” on Page 5 of the revised main text.

Reviewer #2:

In this manuscript, the authors reported the observation of clearly resolved dual pulsed stimulated emission in perovskite multiple quantum wells. The mechanism of double-pulsed stimulated emission has been investigated in detail by using transient spectroscopy measurements and theoretical modelling. Furthermore, the Auger recombination in the perovskite multiple quantum wells in this work behaves obviously different from the normal gain medium. These observations of the ultrashort laser pulse doubling by using optical gain medium are novel and very interesting. The conclusions are convincing, and the manuscript is well-written. I believe this paper will attract extensive attentions in the field of metal-halide perovskites and ultrafast photonics. Therefore, I recommend the publication of this manuscript in Nature Communications.

I have few minor suggestions listed below and hope the authors would consider before publication:

Response: We are delighted to hear the reviewer's recognition on the novelty and importance of our research discovery and greatly appreciate his/her recommendation for publication of our work. The reviewer's critical comments are very helpful for us to further improve the manuscript. We have carefully considered the reviewer's comments and made the necessary revision as follows:

1. There is very little discussion about previous works of the light-induced lattice dynamics on the stimulated emission in perovskites. The coherent acoustic phonon was observed in 2D perovskites [Nat. Commun. 9, 2019 (2018)]. How do the authors reconcile their observations with this previous report?

Response: We thank the reviewer for sharing this reference that we were not aware of. Although the reported periods of coherent acoustic phonon oscillations in 2D perovskites have the time range of tens of ps, however, the coherent acoustic phonon oscillations could be observed with different pump wavelengths in a large pump fluence range. The double-pulsed StE in our work was only observed with above band gap (the bilayer perovskite) photon pumping at high pump fluence above the specific threshold, and no oscillations were observed in our TRPL measurements. Thus, the coherent acoustic phonon oscillation is unlikely to be the origin of double-pulsed StE.

We have added the relevant reference and discussions as shown in the following response to comments 2 in the revised manuscript.

2. Can the authors rule out other possible reasons for the double pulsed stimulated emission? For example, the multiple reflection of the stimulated emission pulse within the film.

Response: We thank the reviewer for the valuable comment.

To double confirm our observed double pulsed stimulated emission from MQWs under 400 nm excitation, we did carefully control measurements as presented in the manuscript. We measured 3D perovskites under 400-nm excitation, and MQWs under 650-nm excitation with the same condition (pump configuration, pump beam spot size, and spot shape), in which only one

stimulated emission pulse was observed. Thus, if the double pulsed stimulated emission is from multiple reflection within the film, it should not dependent on the pump wavelength.

To make our explanation clearer, we thus added the following discussions on page 7 of revised manuscript:

“These observations rule out other possible origins for double-pulsed StE, such as the multiple reflection in the sample³¹, which should not be pumping photon energy dependent. In addition, the exciton-lattice interaction in 2D perovskite QWs^{32, 33} is unlikely to be the origin of double-pulsed StE as no oscillations is observed in our TRPL measurements at different pump fluence. ”

3. Do the authors observe any sample degradation under these conditions, and could this influence the overall carrier dynamics?

Response: We thank the reviewer for the valuable comment. The Ruddlesden-Popper perovskite MQWs are well-recognized because of their high stability. Our measurements of PL, time-resolved PL (TRPL), UV-Vis and transient absorption (TA) in this study all used same samples and were measured within one month with samples kept in the nitrogen atmosphere or vacuum. We did not observe the degradation under the measurements as reflected from our results. For example, the emission positions in PL spectra measured by spectrometer, and transient PL spectra in TRPL measured by streak camera are consistent; the bleaching positions from bilayer QWs are also consistent with transition positions measured by UV-vis; the rising time measured from TRPL peak are consistent with that probed at bleaching peaks in TA measurements. Thus, there should not be any sample degradation during our measurements that could affect the overall carrier dynamics.

4. Can the authors provide more explanation for the shoulder on the right of the second pulse in Fig. 2b at ~170ps at high pump fluence. Is it from the third stimulated emission pulse, why it is much weaker than second pulse?

Response: We thank the reviewer for the valuable comment. We speculate that the weak shoulder on the right of second StE pulse at ~170 ps (Fig. 2b) may originate from the carrier funneling from some farther LD QWs to the gain medium. To make our statement clearer to readers, we have added: “The weak shoulder on the right of the second StE pulse at ~170 ps (Fig. 2b) may originate from the carrier funneling from some farther LD QWs to the gain medium after the second StE, and the weaker intensity may be due to the limited amount of such LD QWs.” on Page 12 in the revised manuscript.

Reviewer #3:

General Comments: *The authors of this manuscript present a study on the amplified spontaneous emission (ASE) in 2D perovskites. The particular structure of the material allows for a charge carrier reservoir, such that, after an initial ASE peak, a second peak is observed at longer time delay, due to funneling of carriers from the reservoir into the emitting state on a picosecond time scale.*

Notwithstanding the technical merits of the manuscript, I do have the following two major concerns:

1. The authors are overselling their study.

-They study concerns the observation of ASE with two time constants. This is not equal to coherent light pulse doubling, at no point the coherence of the light pulses is demonstrated.

Response: We appreciate the reviewer for his/her time to review our manuscript. After careful consideration and reviewing our results (one input ultrashort (fs) pumping laser pulse was doubled into two output ultrashort (ps) laser pulses by MQWs in the manuscript, as schematically show in Fig. 3d), we would like to finetune our original title from:

“Ultrashort Coherent Light Pulse Doubling by Metal-Halide Perovskite Multiple Quantum Wells”
to “Ultrashort **Laser** Pulse Doubling by Metal-Halide Perovskite Multiple Quantum Wells”

We believe that this title would be not only clearer and informative to the readers but also retain the novelty and attractivity of our work.

Additionally, we have replaced the statements from “coherent light pulse” to “**laser** pulse” throughout the main text.

-They also claim to produce ultrashort pulses, yet the pulsed produced are more than two orders of magnitude longer than the 150 fs excitation pulses.

Response: We thank the reviewer for the comment.

The reasons we claim to produce “ultrashort” pulse doubling are (i) the incident (femtosecond) laser pulse is doubled into two (picosecond) laser pulses by the MQWs; (ii) ultrashort pulses are generally defined as laser pulses with durations in the scale of femtoseconds to a few of tens of picoseconds, thus, our observed ps stimulated emission pulses can be considered as a type of ultrashort pulses.

The ps pulse widths are due to the following reasons: (i) no cavity and model lock techniques are used for our gain media. With these techniques, we believe that the pulse width could be greatly reduced. (ii) The observed tens of ps pulse width is limited by the time resolution of ~ 20 ps of our streak camera (which was mentioned on Page 7 of the main text).

As the main objective of this manuscript is to report the observation of the double pulsed stimulated emission and explain the underline photophysics, thus the investigation to further reduce the pulse width is not conducted. And we believe this decision would not affect our conclusions.

2. They also state that their results now enable a miniaturization of devices capable of light pulse doubling. However, on-chip manipulation of light pulses has already been demonstrated in silicon integrated photonics. Typically passive control of the light pulses is achieved, for instance, by microring resonators or photonic crystals acting as an optical delay line (e.g. doi: 10.1109/JPHOT.2010.2044989). Combined with waveguide splitters, one could be able to achieve a pulse train with controllable delay between pulses, in essence what the researchers aim to show here but with a higher degree of control.

Response: We thank the reviewer for the constructive comment. We fully agree with reviewer that on-chip manipulation of light pulses has already been demonstrated in silicon integrated photonics. To address the reviewer’s concern, we have revised the sentences in our abstract and introduction as shown below:

(i) “However, such light pulse doubling, so far, is solely realized with bulky external optical elements, which are inconducive to miniaturization and device integration. Nevertheless, the employment of these external bulky optical setups was inevitable as the ultrashort-laser-pulse number is difficult to be multiplied by the optical gain medium itself.” in the abstract of manuscript has been changed to

“However, **the laser pulse multiplication**, so far, is solely realized with **the optical/mechanical setups or devices to modify the output laser pulses from the gain medium**. Nevertheless, **the employment of these external methods** was inevitable as the ultrashort-laser-pulse number is difficult to be multiplied by the optical gain medium itself.” in revised manuscript.

(ii) And revised “However, the employment of these external bulky optical setups is inconducive to miniaturization and device integration. This limitation stimulates the exploration of whether there is a new type of optical gain medium with internal optical properties that is capable of generating multiple laser pulses by single ultrashort laser pulse excitation.” in the introduction of the manuscript into

“**All these technologies are using external optical/mechanical methods to modify the output laser pulse from the gain medium, which simulate our interest on** whether there is a new type of optical gain medium with **unique optical properties and carrier dynamics** that is capable of generating multiple laser pulses.” on Page 2 of the revised manuscript.

In conclusion, considering the elements above, I find the manuscript more suitable for a more specialized journal, as it doesn’t meet the stringent criteria of novelty and potential impact that merits publication in Nature Communications.

Response: We acknowledge that the significance and novelty of the findings in our original manuscript may not have been adequately highlighted, which could have affected the reviewer's view about our research. To sum up, the significance and novelty of our findings lie in the observation of new photophysics in carrier dynamics in this novel gain material (not new kind of devices). Specifically, *we, for the first time:*

1. Demonstrated the double pulsed stimulated emission from a novel gain medium, which was not observed in normal gain media before.

2. Provided clear explanation of the mechanism of double-pulsed stimulated emission by combining the detailed transient spectroscopy measurements and theoretical modelling.

3. Observed nearly Auger-free stimulated emission in this novel gain medium, which was rarely observed in traditional optical gain media.

We believe that these findings have a profound bearing on our understanding of the photophysics of halide perovskites and have strong implications on their applications as optical gain medium. Moreover, they will also play an important role in providing critical knowledge to guide material processing and device engineering efforts to develop new type of laser pulse manipulating devices. We hope our above justification could satisfy the referee concerns.

REVIEWERS' COMMENTS:

Reviewer #1 (Remarks to the Author):

The revised version of the manuscript has addressed the previously raised questions from all the reviewers. I think the observed phenomenon is very interesting and important for lead halide perovskite research. The physical picture is also complete. This phenomenon, associated with the nanocavity, is able to produce the sub-picosecond laser pulses and has the potential to solve the problem of on-chip integrated coherent light sources. As a result, I am pleased to suggest to publish it on Nature Communications.

Reviewer #2 (Remarks to the Author):

The authors have addressed all my concerns clearly. I believe the revised manuscript is ready for publication.

Reviewer #3 (Remarks to the Author):

The authors have addressed my comments satisfactorily. I have no further comments.

- All reviewer comments are displayed in *dark orange italics*.
- All our responses are displayed in black.

Reviewer #1(Remarks to the Author):

The revised version of the manuscript has addressed the previously raised questions from all the reviewers. I think the observed phenomenon is very interesting and important for lead halide perovskite resaerch. The physical picture is also complete. This phenomenon, associated with the nanocavity, is able to produce the sub-picosecond laser pulses and has the potential to solve the problem of on-chip integrated coherent light sources. As a result, I am pleased to suggest to publish it on Nature Communications.

Response: We thank the reviewer for his time to review and help us strengthen our manuscript. We are delighted to hear the reviewer’s recognition on the novelty and importance of our research discovery and greatly appreciate his/her recommendation for publication of our work.

Reviewer #2 (Remarks to the Author):

The authors have addressed all my concerns clearly. I believe the revised manuscript is ready for publication.

Response: We thank the reviewer for his time to review and help us strengthen our manuscript and greatly appreciate his/her recommendation for publication of our work.

Reviewer #3 (Remarks to the Author):

The authors have addressed my comments satisfactorily. I have no further comments.

Response: We thank the reviewer for his time to review and help us strengthen our manuscript and greatly appreciate his/her recommendation for publication of our work.